# IFITMs from Naturally Infected Animal Species Exhibit Distinct Restriction Capacities against Toscana and Rift Valley Fever Viruses

**DOI:** 10.3390/v15020306

**Published:** 2023-01-22

**Authors:** Marie-Pierre Confort, Maëva Duboeuf, Adrien Thiesson, Léa Pons, Federico Marziali, Sophie Desloire, Maxime Ratinier, Andrea Cimarelli, Frédérick Arnaud

**Affiliations:** 1IVPC UMR754, INRAE, Univ Lyon, Université Claude Bernard Lyon 1, EPHE, PSL University, F-69007 Lyon, France; 2Centre International de Recherche en Infectiologie (CIRI), Université de Lyon, Inserm, U1111, Université Claude Bernard Lyon 1, CNRS, UMR5308, École Nationale Supérieure de Lyon, F-69342 Lyon, France

**Keywords:** Toscana virus, Rift Valley fever virus, IFITM, viral restriction

## Abstract

Rift Valley Fever virus (RVFV) and Toscana virus (TOSV) are two pathogenic arthropod-borne viruses responsible for zoonotic infections in both humans and animals; as such, they represent a growing threat to public and veterinary health. Interferon-induced transmembrane (IFITM) proteins are broad inhibitors of a large panel of viruses belonging to various families and genera. However, little is known on the interplay between RVFV, TOSV, and the IFITM proteins derived from their naturally infected host species. In this study, we investigated the ability of human, bovine, and camel IFITMs to restrict RVFV and TOSV infection. Our results indicated that TOSV was extremely sensitive to inhibition by all the animal IFITMs tested, while RVFV was inhibited by human IFITM-2 and IFITM-3, but not IFITM-1, and exhibited a more heterogeneous resistance phenotype towards the individual bovine and camel IFITMs tested. Overall, our findings shed some light on the complex and differential interplay between two zoonotic viruses and IFITMs from their naturally infected animal species.

## 1. Introduction

Rift Valley fever virus (RVFV) and Toscana virus (TOSV) are members of the *Phenuiviridae* family (*Phlebovirus* genus), the largest and, perhaps, the most diverse family of RNA viruses. These viruses are arthropod-borne and are responsible for zoonoses that affect both humans and/or livestock. Despite the fact that they are closely related, RVFV and TOSV infections lead to very distinct clinical outcomes that largely depend on the host reservoir. TOSV infects a broad range of mammalian species, including humans, sheep, cows, and horses [1,2]. While its infection is asymptomatic in animals, in humans TOSV causes acute—albeit almost always non-fatal—meningitis and/or encephalitis in approximately 30% of cases [3,4,5]. RVFV, on the other hand, has been classified by the World Health Organization in the top ten priority list of emerging pathogens likely to cause severe outbreaks in the future. This phlebovirus is indeed responsible for high rates of morbidity and mortality in animals (essentially sheep and cattle) as well as abortions in pregnant ruminants (sometimes up to 100%), whereas camels are generally asymptomatic RVFV carriers [6,7,8,9,10]. Conversely, in humans, RVFV infections typically lead to self-limiting, acute, and febrile illnesses, with a few patients (1–3%) developing more severe diseases such as fulminant hepatitis associated with hemorrhage and, occasionally, retinitis or late-onset encephalitis, with a fatality rate that ranges from 10% to 20% [11,12,13,14,15]. The pathogenic mechanisms underlying the different outcomes of RVFV and TOSV infections are still unknown, although they are likely modulated by virus characteristics as well as host-specific interactions.

Interferon-induced transmembrane (IFITM) proteins are important restriction factors that interfere with the entry of a wide range of viruses, by blocking viral fusion with host cells and sequestering incoming virions in endosomes [16,17]. IFITMs have also been described to interfere at later stages of the viral cycle by leading to the production of less-infectious viral particles [18,19]. The human genome codes for several IFITMs, including human IFITM-1 (hIFITM-1), human IFITM-2 (hIFITM-2), and human IFITM-3 (hIFITM-3), extensively studied for their antiviral properties. The members of this family share a common topology, consisting of an intramembrane domain (IMD) and a transmembrane domain (TMD) separated by an intracytoplasmic loop (CIL), and a N-terminus domain (NTD) and a C-terminus domain (CTD) of variable lengths, with the first involved in protein subcellular localization. The hIFITM-2 and hIFITM-3 possess shorter CTDs and longer NTDs than hIFITM-1, with two overlapping endocytic signals—PPxY and YXXφ—that deliver them mostly, albeit not exclusively, to the inner membrane of intracellular compartments [20,21,22]. IFITMs’ antiviral abilities have been widely studied in several vertebrates, including humans, mice, chicken, pigs, ducks [23], dogs [24], and bats [25,26,27]. The emerging picture suggests an extensive functional heterogeneity among animal IFITMs with respect to different viruses, thereby highlighting the importance of studying IFITM–virus interplay in natural settings. Along these lines, little is known about the functions of IFITMs in animal species such as bovine and camels, which are naturally infected by RVFV and/or TOSV. RVFV has been described to be restricted by hIFITM-2 and hIFITM-3 (but not by hIFITM-1) at early stages of viral replication [28], whereas the RVFV MP12 strain appears to be resistant to all human IFITMs at late stages of viral replication [19]. 

In this study, we characterized the cellular localization and antiviral activities of several bovine and camel IFITMs, as these hosts are particularly relevant to the ecology of several zoonotic viruses, including RVFV and TOSV. Our findings reveal the differential sensitivity of RVFV and TOSV towards IFITMs, suggesting that virus–IFITM interplay may play a critical role in virus–host coevolution and viral pathogenesis.

## 2. Materials and Methods

### 2.1. IFITM Sequence Searches and Expression Plasmids

Human and bovine IFITMs have been described previously [29]. *Camelus* IFITM sequences were retrieved from the NCBI GenBank database and chemically synthesized (Genewiz, Leipzig, Germany) as an N-terminal HA fusion within the pcDNA3.1 plasmid. Of note, the camel genome is poorly annotated, and its annotations are constantly updated. Given that the IFITM nomenclature tends to be misleading in poorly annotated genomes, bovine and camel IFITMs have been named as a, b, and c (rather than 1, 2, and 3) to avoid possible name-based ambiguity that would suggest that they are direct orthologs of human IFITMs.

### 2.2. IFITM Phylogenetic Analysis and Protein Sequences Alignment

Phylogenetic analyses were performed using MEGA X software version 10.0.4 [30]. Multiple alignments of IFITM nucleotide sequences were created using MUltiple Sequence Comparison by Log-Expectation (MUSCLE) version 3.8.31 [31]. Based on this alignment, a maximum likelihood tree was constructed with the Taruma-3 parameter model with a discrete Gamma distribution [32]. The robustness of the nodes was tested by 1000 bootstrap replications. A protein alignment was also generated using the MUltiple Sequence Comparison (MUSCLE) software.

### 2.3. Cell Cultures

VeroE6, HEK293T, and BSR cells were grown in Dulbecco’s Modified Eagle Medium (DMEM; Gibco, Thermo Fisher Scientific, Villebon-sur-Yvette, France), supplemented with 10% heat-inactivated foetal bovine serum (FBS; GE HEALTHCARE Europe GmbH, Freiburg, Germany), and 25 µg/mL penicillin-streptomycin (p/s; Gibco, Thermo Fisher Scientific). All the cell lines were cultured in a 37 °C, 5% CO_2_ humidified incubator. VeroE6 and HEK293T cells were purchased from ATCC. BSR cells were kindly provided by Prof Karl Conzelmann (Ludwig-Maximilians University Munich—Gene Center—Munich/Germany).

### 2.4. Immunofluorescence and Confocal Microscopy Experiments

HEK293T cells were directly seeded onto glass coverslips coated with 0.01% poly-L-Lysine. Cells were transfected with plasmids, coding the indicated IFITMs with Lipofectamine 3000 according to the manufacturer’s instructions (Thermo Fisher Scientific, Villebon-sur-Yvette, France). Twenty-four hours later, cells were fixed with 3.7% paraformaldehyde (PFA; CliniSciences, Nanterre, France) and permeabilized with 0.1% Triton (Euromedex, Souffelweyersheim, France) in phosphate buffered saline (PBS; Sigma-Aldrich, Merck, Saint-Quentin Fallavier, France), prior to overnight incubation with 1:100 anti-HA mouse monoclonal antibody (H3663, Sigma-Aldrich, Merck, Saint-Quentin Fallavier, France), followed by 1 h incubation with 1:1000 secondary antibody conjugated to Alexa488 (A21202, Thermo Fisher Scientific, Villebon-sur-Yvette, France). Coverslips were stained with a DAPI solution diluted 1:10,000 in PBS, and mounted using the anti-quenching Fluormount G (Southern Biotech, Birmingham, AL, USA). Images were collected using a Zeiss LSM 880 AiryScan confocal microscope, and pictures were analyzed with ImageJ software version 1.53t.

### 2.5. Virus Stocks and Titrations

The RVFV ZH548 strain was amplified on VeroE6 cells for 3 days in DMEM supplemented with 4% FBS and 1% p/s. Virus titers were determined by standard plaque assays using VeroE6 cells. Briefly, monolayer VeroE6 cells seeded onto 12-well plates were infected for 2 h at 37 °C with 10-fold dilutions of the virus stocks. After removing the medium containing the virus, the cells were washed twice with PBS. Next, the cells were covered with 1.5 mL of a semisolid overlay containing 3% ultrapure agarose (Life Technologies, Thermo Fisher Scientific, Villebon-sur-Yvette, France) in minimal essential medium (MEM; Gibco, Thermo Fisher Scientific, Villebon-sur-Yvette, France) supplemented with 2% FBS and 1% p/s, and incubated at 37 °C for 5 days. Finally, the cells were fixed in 4% formaldehyde (Fisher Scientific, Illkirch, France) overnight at 4 °C, the agarose overlay was removed, and the cells were washed with PBS and stained with 0.2% crystal violet, 3.7% formaldehyde, and 20% ethanol solution (Sigma-Aldrich, Merck, Saint-Quentin Fallavier, France). TOSV strain MRS2010-4319501 (EVAg, Marseille, France) was amplified on BSR cells for 6 days in DMEM supplemented with 4% FBS and 1% p/s. Virus titers were determined by standard plaque assays using VeroE6 cells. The cells were infected and washed as described above. Next, the cells were covered with 1.5 mL of a semisolid overlay containing 3.2% carboxymethyl cellulose (Sigma-Aldrich, Merck, Saint-Quentin Fallavier, France) in MEM supplemented with 2% FBS and 1% p/s, and then incubated at 37 °C for 6 days. Finally, the overlay was discarded; the cells were washed with PBS, and plates were fixed in 4% PFA overnight at 4 °C. The plates were then stained as described for RVFV titration. Viral titers were expressed as plaque-forming units (PFU/mL). 

### 2.6. Transfection and Infection Assays

2.5 × 10^5^ HEK293T cells were plated onto 12-well plates. The following day, they were transfected with 500 ng of pcDNA3.1 or different IFITM expression plasmids, named as follows: pcDNA-hIFITM-1, pcDNA-hIFITM-2, pcDNA-hIFITM-3, pcDNA-bIFITM-a and pcDNA-bIFITM-b (Bovine), pcDNA-cIFITM-a, pcDNA-cIFITM-b, or pcDNA-cIFITM-c (Camel), with 2 µL Lipofectamine 2000 (Invitrogen by life technologies, Thermo Fisher Scientific, Villebon-sur-Yvette, France) in a final volume of 100 µL of OptiMEM (Gibco, Thermo Fisher Scientific, Villebon-sur-Yvette, France). After incubation for 6 h at 37 °C and 5% CO_2_, 1 ml of DMEM supplemented with 4% FBS and 1% p/s was added to each well, and the cells were further incubated at 37 °C. Twenty-four hours post-transfection, cells were infected either with the ZH548 RVFV strain or the MRS2010-4319501 TOSV strain at a multiplicity of infection (MOI) of 0.001, in 500 µL DMEM supplemented with 4% FBS and 1% p/s. After 2 h at 37 °C, the virus inocula were removed, the cells were washed once with PBS prior to adding 600 µL of DMEM supplemented with 4% FBS and 1% p/s. Twenty-four hours post-infection, the cell supernatants were collected for viral titration by limiting dilution assays in BSR cells. Briefly, in a 96-well plate, 11 µL of viral supernatant were diluted in quadruplicate by a 10-fold serial dilution (from 10^−1^ to 10^−12^) in 100 µL of DMEM supplemented with 4% FBS and 1% p/s. Then, 8 × 10^3^ BSR cells were added to each well and incubated for 5 days at 37 °C in 5% CO_2_. Cytopathic effects were assessed under a microscope, and viral titers were expressed as 50% tissue culture infective doses (TCID50/mL) using the Reed and Muench method [33]. Statistical analyses were performed using R and RStudio software. The Kruskal–Wallis test was first used with a significance threshold of 0.01. In case of statistical significance, multiple Wilcoxon–Mann–Whitney tests were used as follow-up tests, with pcDNA3.1 as the reference group and *p*-value adjustment [32]. Cells were then either collected for Western blotting after cell lysis with 200 µL 1× Laemmli buffer (Alfa Aesar™, Fisher Scientific, Illkirch, France) and heat treated at 95 °C for 10 min, or for flow cytometry analysis by detaching them with 1× trypsin–EDTA 0.05% (200 μL, Gibco, Thermo Fisher Scientific, Villebon-sur-Yvette, France) and then fixing them in 400 µL 4% PFA. Experiments were performed in triplicate and at least three times, independently, using two different virus stocks.

### 2.7. Western Blotting

Proteins were separated by SDS-PAGE on 12% polyacrylamide gel and transferred to nitrocellulose membranes (Trans Blot Turbo RTA transfer kit midi, BioRad, Marnes-La-Coquette, France). After a blocking step in tris buffer saline (TBS) with 0.1% Tween and 5% dry milk (Euromedex, Souffelweyersheim, France), membranes were incubated overnight at 4 °C with a polyclonal rabbit antibody against RVFV N (1:5000) or TOSV N (1:2000), or a monoclonal mouse antibody against HA (1:5000; Ab130275, Abcam, Amsterdam, The Netherlands). Actin was detected with monoclonal mouse antibody coupled with horseradish peroxidase (HRP; 1:75,000; A3854, Sigma-Aldrich, Merck, Saint-Quentin Fallavier, France). After washing once with a solution of TBS with 0.1% Tween, membranes were exposed to the appropriate peroxidase-conjugated secondary antibodies (1:5000 goat anti-rabbit IgG A6154 or 1:10,000 sheep anti-mouse IgG A5906; Sigma-Aldrich, MERCK) for 1.5 h, washed, and then visualized by chemiluminescence using the Clarity ECL Western blotting substrate (BioRad) and ChemiDoc XRS+ System (BioRad). Each experiment was performed at least three times, independently. TOSV and RVFV N protein, HA-IFITMs, and actin signals were quantified using Image Lab software (Bio-Rad). Statistical analyses were performed with the Kruskal–Wallis test using NCSS9 software. The polyclonal rabbit antibodies against RVFV N and TOSV N were kindly provided by Dr Benjamin Brennan and Prof. Alain Kohl (MRC—University of Glasgow Centre for Virus Research—Glasgow), respectively [34,35].

### 2.8. Flow Cytometry

Fixed cells were permeabilized using PBS with 0.1% Saponin (Sigma-Aldrich, Merck, Saint-Quentin Fallavier, France) and 10% FBS. TOSV-infected cells were detected using mouse ascites against TOSV (1:1000), kindly provided by Dr. Philippe Marianneau, (ANSES de Lyon, France). RVFV-infected cells were detected using a polyclonal rabbit antibody raised against RVFV N (1:1000), while IFITM was detected using a monoclonal mouse antibody raised against HA (1:1000). Secondary antibodies were goat anti-rabbit IgG conjugated with Alexa Fluor 488 (1:1000; A11034, ThermoFisher Scientific, Villebon-sur-Yvette, France) and goat anti-mouse IgG2b conjugated with PE/Cy7 (1:1000; ab130790, Abcam). The stained cells were analysed by flow cytometry using an LSR II machine (Becton Dickinson BD, Le Pont-de-Claix, France). Data acquisition and analysis were performed using BD FACS Diva software (Becton Dickinson BD). The values obtained by control cells (i.e., cells infected with the pcDNA3.1 plasmid) were arbitrarily set as 100%, and compared to those of cells transfected with the different IFITMs. Statistical analyses were performed using R and RStudio software. The Kruskal–Wallis test was first used with a significance threshold of 0.01. In case of statistical significance, multiple Wilcoxon–Mann–Whitney tests were used as follow-up tests, with pcDNA3.1 as the reference group and *p*-value adjustment [36].

## 3. Results

### 3.1. Characterization of Human, Bovine, and Camel IFITMs

From the sequence searches in the NCBI GenBank database, we identified three camel IFITM sequences: cIFITM1L (*Camelus dromedarius*, XM_011000007.1), cIFITM1 (*Camelus dromedarius*, XM_010999546.1), and cIFITM3 (*Camelus ferus*, XM_032490416.1). Animal IFITMs are generally named analogously to their human counterparts, depending on the length of their variable N- and C-terminus. This nomenclature may be misleading, especially in poorly annotated genomes and in IFITMs that exhibit different combinations of N- and C-terminus lengths than human IFITMs. In the absence of a formal and complete evolutionary analysis, this nomenclature appears misleading, as it may suggest that animal and human IFITMs sharing the same numbers are ortholog pairs. Thus, to avoid possible confusion, we have renamed the bovine and camel IFITMs as a, b, and c, and compared them with each other. 

We found that two camel IFITMs (cIFITM-a and cIFITM-b) possessed a short NTD and, therefore, lacked the YXXφ endocytic sorting motif that is typical of hIFITM-1 (Figure 1), and exhibited a more variable C-terminus. In contrast, the third camel IFITM (cIFITM-c) presented a long NTD and CTD. The two bovine IFITMs tested in this study (bIFITM-a and bIFITM-b) differed in the length of their NTDs, while they possessed CTDs of similar lengths. We also performed a phylogenetic analysis of all the IFITMs, and observed that those from the same species clustered together (Figure 2). 

Next, we investigated the cellular localization of human, bovine, and camel IFITMs in HEK293T cells. As expected, hIFITM-1 exhibited an essentially plasma membrane distribution as opposed to hIFITM-2 and hIFITM-3, which were more classically internally distributed. Several animal IFITMs, including bIFITM-a, cIFITM-a, and cIFITM-b, displayed a plasma membrane distribution resembling that of hIFITM-1. Conversely, bIFITM-b and cIFITM-c were primarily localized in intracellular compartments like hIFITM-2 and hIFITM-3 (Figure 3).

### 3.2. IFITMs from Naturally Infected Species Differentially Restrict TOSV and RVFV

We then studied the ability of human, bovine, and camel IFITMs to inhibit TOSV and RVFV replication. To this end, we transfected HEK293T cells with human, bovine, or camel expression plasmids for IFITMs or with pcDNA as a control. Twenty-four hours post-transfection, we infected these cells with TOSV or RVFV and compared the amount of viral Nucleocapsid proteins (N) in transfected cells. We found that TOSV N protein levels were significantly decreased in the presence of all the IFITMs (*p* < 0.0001; Kruska–Wallis test), and more potently so by hIFITM-2, bIFITM-a, and bIFITM-b (Figure 4A,C). 

Similarly, we observed that all the IFITMs, with the exception of hIFITM-1, were able to reduce RVFV N protein levels, albeit with differences between hIFITM-2, hIFITM-3, bIFITM-b, cIFITM-b, and cIFITM-c that all displayed a higher restriction effect with higher statistical significance (*p* < 0.01; Kruskal–Wallis test) than bIFITM-a and cIFITM-a (*p* = 0.024 and *p* = 0.013, respectively). Interestingly, and unlike other IFITMs, cIFITM-c strongly decreased RVFV N protein to barely detectable levels, while hIFITM-1 significantly increased it (*p* < 0.001; Figure 4B,D).

To further confirm these data, we conducted flow cytometry experiments on TOSV and RVFV-infected cells and then measured the viral titers of the corresponding cell supernatants. We found that all the IFITMs reduced the number of TOSV-infected cells by approximately 50% compared to the pcDNA control (Figure 5A). Conversely, following RVFV challenge, we found that, while hIFITM-2, hIFITM-3, bIFITM-b, and cIFITM-c reduced the number of infected cells by about 50%, bIFITM-a and cIFITM-a showed very little to no restriction activity and hIFITM-1 significantly promoted the percentage of RVFV-infected cells (Figure 5B). Interestingly, we found that, unlike cIFITM-a, cIFITM-b significantly decreased the number of RVFV-infected cells (*p* < 0.001; Wilcoxon–Mann–Whitney test). Importantly, we found comparable numbers of IFITM-positive cells in the different conditions tested (between 30% and 50%), and we did not observe any significant difference in the number of human and bovine IFITM-expressing cells between the TOSV and RVFV experiments (*p* > 0.01; Kruskal–Wallis test; Figure 5C,D).

Finally, we found that all the IFITMs, with the exception of hIFITM-1, significantly reduced TOSV viral titers in the cell supernatants (Figure 6A). Conversely, when infecting with RVFV, all the IFITMs tested, except hIFITM-1, bIFITM-a, and cIFITM-a, led to a significant decrease in viral infectious titers (Figure 6B). 

## 4. Discussion

In this study, we demonstrated that TOSV and RVFV, members of the same *Phlebovirus* genus, present differential sensitivities towards IFITM-mediated restriction activity. More specifically, we found that TOSV infection was restricted in the presence of all the human and bovine IFITMs tested, indicating an exquisite susceptibility to these restriction factors. Conversely, we found that although the more pathogenic RVFV was still strongly blocked by certain IFITMs, it could resist inhibition by three distinct IFITMs: hIFITM-1, bIFITM-a, and cIFITM-a. Overall, our data indicate that closely related viruses, with similar viral cycles, can be differently affected by IFITMs. Moreover, and most importantly, our results demonstrate that animal IFITMs exhibit important differences in their ability to restrict related viruses. The reason underlying these distinct viral responses is still unclear, and may be either due to their divergence in terms of viral sequences, or to the ability of these viruses to code for efficient viral antagonists against some of these IFITMs. Another possibility could be that IFITMs differentially restrict these two viruses either at early stages—by blocking viral entry—, and/or at late stages—by reducing the infectivity of newly produced virions. However, our experimental settings did not allow us to discriminate between these two modes of action. Hence, it would be interesting to investigate the ability of each IFITM to induce an early and/or late restriction of TOSV and RVFV infection. Deciphering the mechanism(s) undertaken by certain IFITMs to exert distinct (sometimes even opposite) antiviral activities over RVFV will certainly provide important information on phlebovirus–IFITM interplay. 

Interestingly, we observed that cIFITM-b and cIFITM-c successfully inhibited RVFV infection and mostly localized at the plasma membrane and intracellular compartments of transfected cells, respectively. We could speculate that this differential subcellular localization may allow cells expressing both cIFITM-b and cIFITM-c to induce a synergistic inhibition at different stages of its replication cycle. Moreover, we found that, if we exclude their NTDs from the multiple sequence alignment, cIFITM-b and cIFITM-c share 94% protein homology. Hence, we could hypothesize that the determinants of the restriction activity of these proteins are most probably located in their central region and/or CTD. Moreover, we found that cIFITM-c reduced RVFV infectious titers more efficiently than cIFITM-b, suggesting that its NTD is not necessary but may somehow promote and/or increase its viral restriction activity, possibly at IFITM late restriction, thereby reducing virion infectivity.

Finally, we observed that hIFITM-1 did not restrict RVFV infection, in accordance with previous findings [28]. However, we also showed, for the first time, that hIFITM-1 promotes RVFV infection. To date, only a few IFITMs have been shown to act this way, including human and mouse IFITM2 and IFITM3, both able to stimulate human coronavirus OC43 infection [37]. Interestingly, it was recently reported that the endogenous expression (i.e., not overexpression) of human IFITM proteins promotes the efficient infection of human cells by SARS-CoV-2 [38]. Therefore, we could hypothesize that, in our experimental condition, the level of hIFITM-1 proteins expressed was suitable for promoting RVFV infection. Overall, all these findings support the hypothesis that the ability of IFITMs to block or promote viral infections is tightly regulated, and also depends on the virus considered.

In summary, we demonstrated that TOSV is restricted by a larger panel of IFITMs compared to RVFV. Indeed, while we observed that TOSV is effectively inhibited by all the human and bovine IFITMs tested in this study, RVFV is not restricted by several animal IFITMs, which may partially contribute to its higher pathogenicity. It is therefore critical to adopt comparative approaches to understand the complex interplay between IFITMs from different species and related viruses.

## Figures and Tables

**Figure 1 viruses-15-00306-f001:**
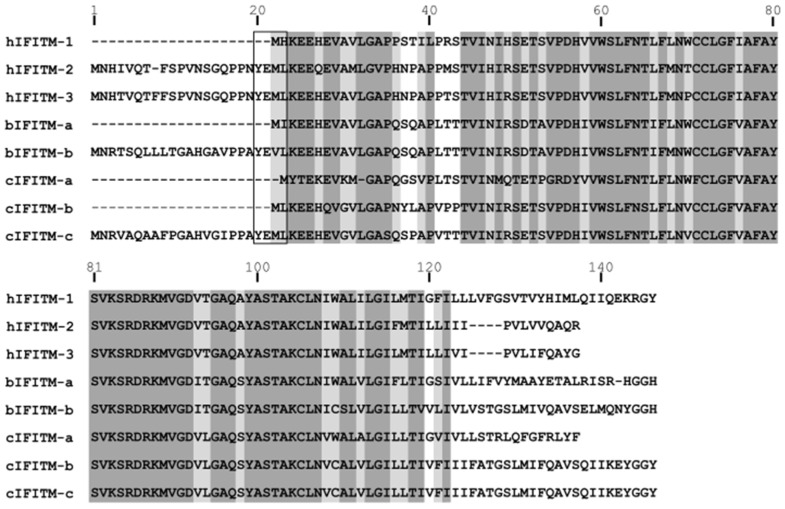
Sequence alignment of human, bovine, and camel IFITMs. The alignment was performed using MUltiple Sequence Comparison (MUSCLE) software. Aligned amino acids are highlighted in dark or light grey depending on whether the residue is present in at least 80% or 50% of the sequences, respectively. Note that shorter N-terminal domains are characterized by the absence of the YXXφ motif (rectangle) present in hIFITM-3 as well as in other IFITMs. hIFITM-1, human IFITM-1; hIFITM-2, human IFITM-2; hIFITM-3, human IFITM-3; bIFITM-a, bovine IFITM-a; bIFITM-b, bovine IFITM-b; cIFITM-a, camel IFITM-a; cIFITM-b, camel IFITM-b; cIFITM-c, camel IFITM-c.

**Figure 2 viruses-15-00306-f002:**
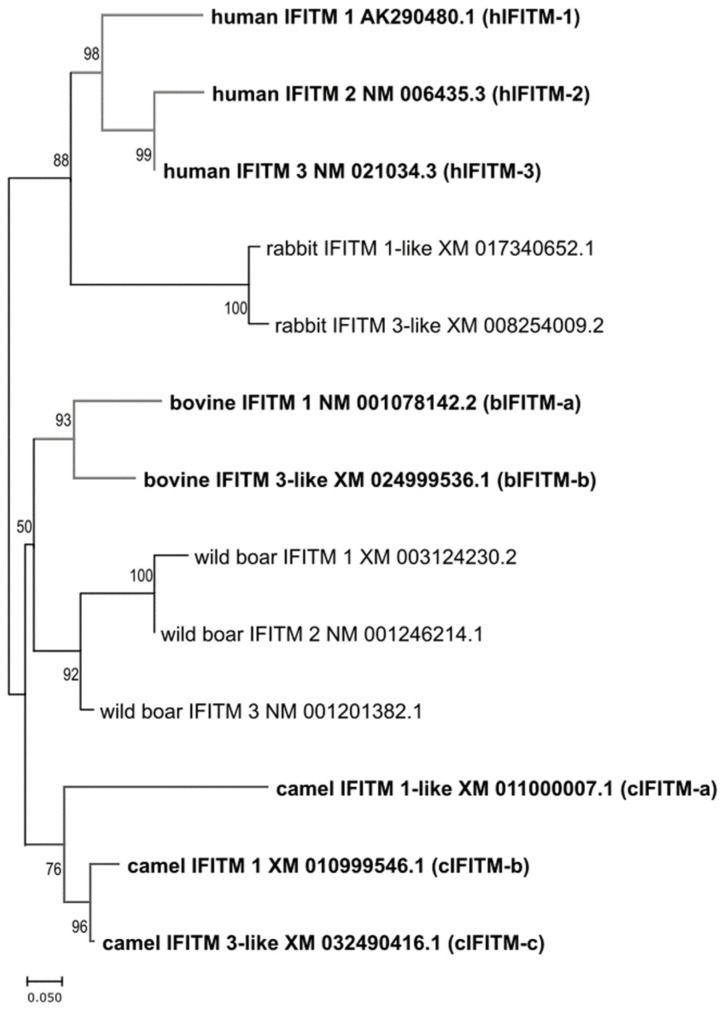
Phylogenetic tree of human, rabbit, bovine, wild boar, and camel IFITMs. The phylogenetic tree was built using the Maximum Likelihood method and Tamura-Nei model with MEGA X software. The percentage of trees in which the associated taxa clustered together is shown next to the branches. In brackets, the names we assigned to the IFITMs analyzed in this study. hIFITM-1, human IFITM-1; hIFITM-2, human IFITM-2; hIFITM-3, human IFITM-3; bIFITM-a, bovine IFITM-a; bIFITM-b, bovine IFITM-b; cIFITM-a, camel IFITM-a; cIFITM-b, camel IFITM-b; cIFITM-c, camel IFITM-c.

**Figure 3 viruses-15-00306-f003:**
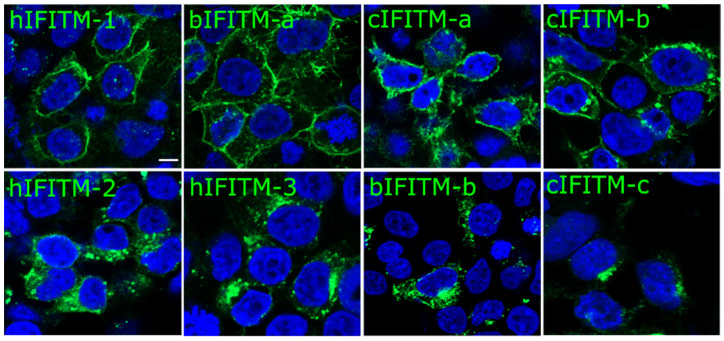
Cellular localization of human, bovine, and camel IFITMs, as indicated. HEK293T cells were transfected with plasmids encoding the different IFITMs. Twenty-four hours post-transfection, the cells were fixed with PFA and immuno-stained for IFITMs with an anti-HA antibody followed by a secondary antibody conjugated to Alexa488 (Green). Cell nuclei were labelled with DAPI staining (blue). Representative pictures of IFITMs expressing cells are shown. The scale bar in the upper left panel represents 5 µm. hIFITM-1, human IFITM-1; hIFITM-2, human IFITM-2; hIFITM-3, human IFITM-3; bIFITM-a, bovine IFITM-a; bIFITM-b, bovine IFITM-b; cIFITM-a, camel IFITM-a; cIFITM-b, camel IFITM-b; cIFITM-c, camel IFITM-c.

**Figure 4 viruses-15-00306-f004:**
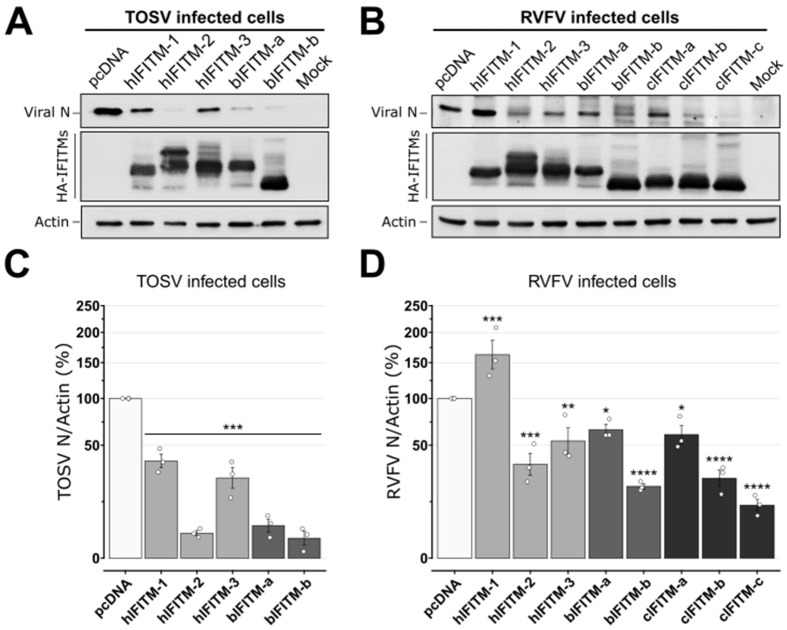
Western blot analyses of TOSV and RVFV N proteins from infected cells expressing the different IFITMs. Infection assays with TOSV (**A**,**C**) or RVFV (**B**,**D**) were performed in HEK293T cells expressing or lacking (pcDNA3.1) the different IFITMs, as indicated. Cells were collected at twenty-four hours post-infection and analyzed by Western blot with antisera raised against TOSV or RVFV N proteins, HA (to detect IFITMs), or actin, as specified. Each experiment was performed three times, independently. Representative blots are presented in panels (**A**,**B**). Signals of TOSV and RVFV N proteins, and actin proteins were quantified from, at least, three independent experiments using Image Lab software. The ratios of TOSV N/Actin (**C**) and RVFV N/Actin (**D**) are reported on the graphs and are relative to those obtained in control cells (pcDNA), arbitrarily set as 100%. Statistical analyses were performed with a Kruskal–Wallis test, and significance (compared to pcDNA control) is presented as follows: *p* < 0.05 (*), *p* < 0.01 (**), *p* < 0.001 (***) and *p* < 0.0001 (****). hIFITM-1, human IFITM-1; hIFITM-2, human IFITM-2; hIFITM-3, human IFITM-3; bIFITM-a, bovine IFITM-a; bIFITM-b, bovine IFITM-b; cIFITM-a, camel IFITM-a; cIFITM-b, camel IFITM-b; cIFITM-c, camel IFITM-c.

**Figure 5 viruses-15-00306-f005:**
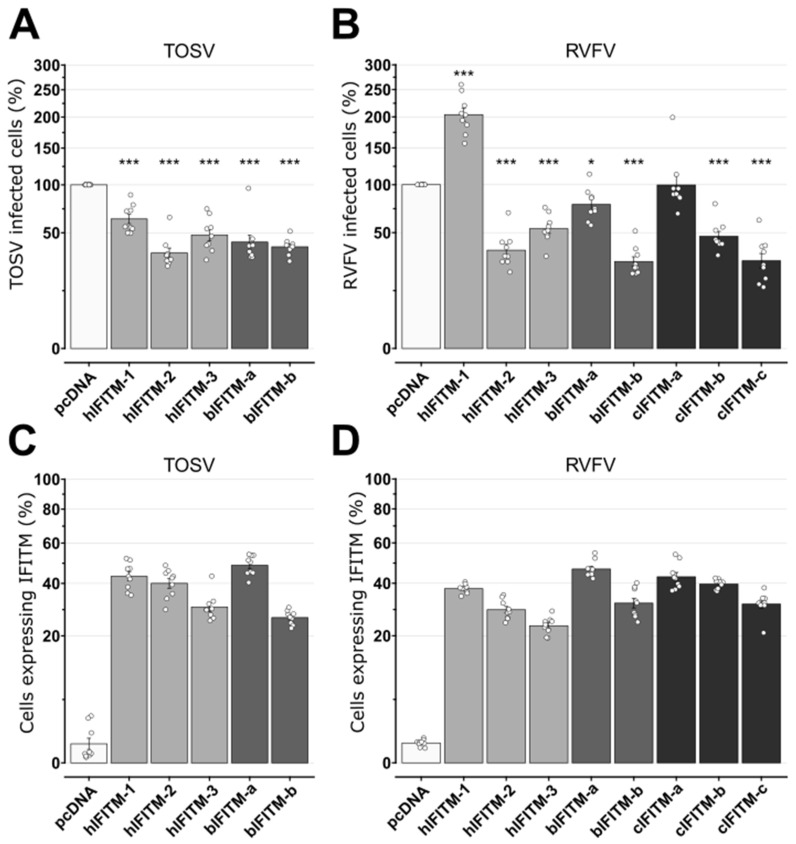
Number of TOSV and RVFV-infected cells expressing the different IFITMs. Infection assays with TOSV (**A**) or RVFV (**B**) were performed in HEK293T cells expressing or lacking (pcDNA) the different IFITMs, as indicated. At twenty-four hours post-infection, cells were collected, fixed in PFA, and immunostained with mouse ascites against TOSV N or a polyclonal rabbit antibody against RVFV N, before being analyzed by flow cytometry. The values in the graphs represent the number of TOSV (**A**) or RVFV (**B**)-positive cells and are relative to those obtained in control cells (pcDNA), arbitrarily set as 100%. Wilcoxon–Mann–Whitney statistical analyses were performed, and significance (compared to pcDNA) is presented as follows: *p* < 0.05 (*) and *p* < 0.001 (***). The percentage of IFITM-expressing cells was also determined by immunostaining using polyclonal rabbit or monoclonal mouse antibodies against HA and flow cytometry analyses in TOSV (**C**) and RVFV (**D**)-infected cells. Each experiment was performed in triplicate and at least three times, independently. Bars indicate standard errors. hIFITM-1, human IFITM-1; hIFITM-2, human IFITM-2; hIFITM-3, human IFITM-3; bIFITM-a, bovine IFITM-a; bIFITM-b, bovine IFITM-b; cIFITM-a, camel IFITM-a; cIFITM-b, camel IFITM-b; cIFITM-c, camel IFITM-c.

**Figure 6 viruses-15-00306-f006:**
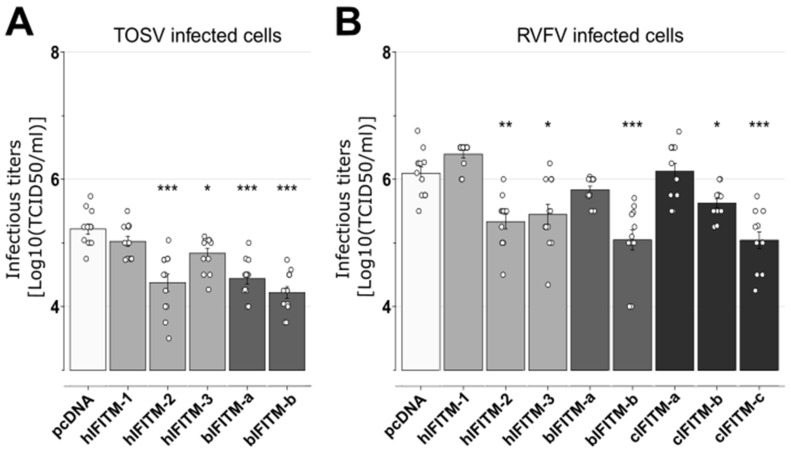
Viral titers of TOSV and RVFV in the presence of human, bovine, or camel IFITMs. Infection assays with TOSV (**A**) or RVFV (**B**) were performed in HEK293T cells expressing or lacking (pcDNA) the different IFITMs, as indicated. At 24 h post-infection, cell supernatants were collected, and viral titers were obtained by limiting-dilution assays in BSR cells. Each experiment was performed in triplicate and at least three times, independently. Bars indicate standard errors. Wilcoxon–Mann–Whitney statistical analyses were performed, and significance (compared to pcDNA data) is presented as follows: *p* < 0.05 (*), *p* < 0.01 (**) and *p* < 0.001 (***). hIFITM-1, human IFITM-1; hIFITM-2, human IFITM-2; hIFITM-3, human IFITM-3; bIFITM-a, bovine IFITM-a; bIFITM-b, bovine IFITM-b; cIFITM-a, camel IFITM-a; cIFITM-b, camel IFITM-b; cIFITM-c, camel IFITM-c.

## Data Availability

The data that support the findings of this study are available upon request.

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
