# Peer review of "IFITMs from Naturally Infected Animal Species Exhibit Distinct Restriction Capacities against Toscana and Rift Valley Fever Viruses"

_viruses, 2023, doi:10.3390/v15020306_

Round 1
Reviewer 1 Report (Previous Reviewer 2)
Overall, the manuscript appears descriptive in nature and could be improved by going deeper into a mechanistic/focused investigation.
Author Response
Please see the attachement file

Reviewer 2 Report (Previous Reviewer 1)
The authors have adequately addressed my concerns.
Reviewer 3 Report (New Reviewer)
This manuscript is quite interesting that clarifies the putative livestock' IFITMs in counteracting the two pathogenic zoonotic bunyaviruses, Rift Valley Fever virus (RVFV) and Toscana virus (TOSV). Although diffs in sequence and evolutionary path, the tested bovine and camel IFITMs have similar inhibitory spectrum with humans, suggesting the conservatory in mammals in counteracting the viral pathogens. The overall science is quite good and logical.
One minor point:
The main conclusion indexed in the abstract may have a broader readership in the scientific community.
Author Response
Please see attachement file

This manuscript is a resubmission of an earlier submission. The following is a list of the peer review reports and author responses from that submission.
Round 1
Reviewer 1 Report
The authors describe data comparing the antiviral activity of human IFITM proteins and some IFITM proteins expressed in bovine and camel species, for which there is relatively little known. The focus on bovine and camel IFITMs is both the greatest strength and greatest weakness of this study for the following reason: there are fundamental problems with the nomenclature used for the IFITM genes used in this study.
Major:
The authors assign names to the bovine and camel IFITMs as if they were orthologs of the human IFITM genes. The problem is that they are not. The authors’ own phylogenetic analysis suggests that, for example, camel “IFITM1” does not share a recent common ancestor with human IFITM1. If they did, you would see that human and camel IFITM1 form a clade. Instead, the authors show that the IFITMs from each species form their own respective clades. This means that the camel IFITMs are more closely related to each other than they are to IFITMs from other species. This suggests that the IFITM locus has expanded independently through gene duplication in each species. As a result, it is not appropriate to call the IFITM genes in bovine or camel “IFITM1-3” because that implies they are direct orthologs with human IFITM genes. Again, they are not. If the authors strive to formally determine whether genes are orthologs, they need to include gene synteny (position in chromosome relative to neighboring genes) as part of their genomic analysis. Simply calling a gene IFITM1, 2, or 3 because the NCBI entry uses this nomenclature, is inappropriate. In the same vein, the lack of a given IFITM1, 2, or 3 in NCBI does not mean that a given species does not have those genes. The IFITM locus must be examined directly and phylogeny and synteny must be used to establish the relatedness of IFITM genes. If the authors want additional evidence supporting that bovine and camel IFITM genes are not orthologous with human IFITM, simply look at the protein alignment provided in Figure 1. The authors seem to be calling IFITM genes 1, 2, or 3 according to the length of the N-terminus. But they are disregarding the C-terminus and other codons that are used as part of the phylogenetic analysis.
The authors should recognize this issue by simply calling these genes something else and rewrite their manuscript accordingly. For example, instead of C1-1, C1-2, and C3, the authors could say Ca, Cb, Cc. Doing so will help them form the proper conclusions and the discussion will be more logical. Right now, for example, the authors try to understand on the sequence level why human IFITM1 and camel IFITM1 do not display the same antiviral activity against RVFV. If the authors conceded that camel IFITM1 is not an ortholog of human IFITM1, this discussion point becomes moot.
Other than this major issue, I do not have any problems with the way the antiviral assays were performed and presented. It is a rather simple and straightforward presentation of data that incrementally adds to our understanding of how animal IFITM works in nature.
Minor:
Remove city and country of origin for the reagents listed in the Methods section.
Reviewer 2 Report
Marie-Pierre Confort et al tested the ability of multiple paralogues of IFITM from human, bovine, and camel to restrict RVFV and TOSV infection. Their result indicated that TOSV is blocked by many IFITMs, including human and bovine IFITM1, IFITM2, and IFITM3. but RVFV is inhibited only by human, bovine, and camel IFITM2 and/or IFITM3, as well as one camel IFITM1. Their finding is incremental to our understanding of IFITM antiviral function. However, the work does not provide much mechanism study and views. Overall, the manuscript appears descriptive in nature and could be improved by going deeper into a mechanistic/focused investigation.
Major points:
1. The authors only examined the IFITMs antiviral activity of inhibiting TOSV and RVFV by measuring infection rate and viral titer, but no mechanistic experiments were carried out, for example, susceptibility of viral entry mediated by envelop glycoprotein from TOSV and RVFV to IFITM should be examined.
2. No mechanistic experiments were carried out to figure out the differential ability of IFITMs to inhibit RVFV and TOSV infection.